# Human Placental Lactogen in Relation to Maternal Metabolic Health and Fetal Outcomes: A Systematic Review and Meta-Analysis

**DOI:** 10.3390/ijms232415621

**Published:** 2022-12-09

**Authors:** Kate Rassie, Rinky Giri, Anju E. Joham, Helena Teede, Aya Mousa

**Affiliations:** 1Monash Centre for Health Research and Implementation (MCHRI), School of Public Health and Preventive Medicine, Monash University, Level 1, 43-51 Kanooka Grove, Clayton, Melbourne, VIC 3168, Australia; 2Department of Diabetes, Monash Health, 246 Clayton Rd, Clayton, Melbourne, VIC 3168, Australia

**Keywords:** birthweight, gestational diabetes mellitus, human chorionic somatomammotropin, human placental lactogen, type 1 diabetes mellitus

## Abstract

Human placental lactogen (hPL) is a placental hormone which appears to have key metabolic functions in pregnancy. Preclinical studies have putatively linked hPL to maternal and fetal outcomes, yet—despite human observational data spanning several decades—evidence on the role and importance of this hormone remains disparate and conflicting. We aimed to explore (via systematic review and meta-analysis) the relationship between hPL levels, maternal pre-existing and gestational metabolic conditions, and fetal growth. MEDLINE via OVID, CINAHL plus, and Embase were searched from inception through 9 May 2022. Eligible studies included women who were pregnant or up to 12 months post-partum, and reported at least one endogenous maternal serum hPL level during pregnancy in relation to pre-specified metabolic outcomes. Two independent reviewers extracted data. Meta-analysis was conducted where possible; for other outcomes narrative synthesis was performed. 35 studies met eligibility criteria. No relationship was noted between hPL and gestational diabetes status. In type 1 diabetes mellitus, hPL levels appeared lower in early pregnancy (possibly reflecting delayed placental development) and higher in late pregnancy (possibly reflecting increased placental mass). Limited data were found in other pre-existing metabolic conditions. Levels of hPL appear to be positively related to placental mass and infant birthweight in pregnancies affected by maternal diabetes. The relationship between hPL, a purported pregnancy metabolic hormone, and maternal metabolism in human pregnancy is complex and remains unclear. This antenatal biomarker may offer value, but future studies in well-defined contemporary populations are required.

## 1. Introduction

Human pregnancy is defined by complex hormonal and metabolic changes, which are essential to regulate nutrient availability and ensure the health of the mother and the growing fetus. In late gestation, a significant increase in maternal insulin resistance (prioritising the delivery of glucose and amino acids across the placenta for use by the fetus) is paralleled by a compensatory increase in insulin synthesis and secretion. The endocrine mechanisms underlying these changes are incompletely understood.

In women with obesity, pre-gestational diabetes (type 1 or 2), or polycystic ovary syndrome (PCOS); the physiological adaptations of pregnancy exacerbate the existing states of insulin resistance and/or deficiency underpinning these conditions. During pregnancy, gestational diabetes mellitus (GDM), defined as carbohydrate intolerance of variable severity with first onset or recognition during pregnancy, is also increasingly common. GDM reflects a failure to sufficiently augment insulin secretion in the face of progressive gestational insulin resistance. Maternal diabetes of any type in pregnancy increases the risk of fetal macrosomia and obstetric complications, and is associated with potential adverse long-term alterations to the metabolic profile of the mother and offspring [1]. GDM during pregnancy is a significant risk factor for future cardiovascular disease in women—one recent meta-analysis of observational trials suggested that women with a history of GDM had a twofold higher risk of future cardiovascular events compared with those who did not [2]. As such, an improved understanding of the mechanisms that alter maternal insulin resistance—and further insights into biomarkers which can facilitate early identification of women at risk of GDM—are key priorities.

Human placental lactogen (hPL), previously known as human chorionic somatomammotropin, is a polypeptide hormone produced during pregnancy by the syncytiotrophoblast cells of the placenta. A member of the somatotropin family, hPL is structurally homologous to pituitary growth hormone (GH), prolactin (PRL) and placental growth hormone (GH-V) [3]. In humans, hPL binds mainly to the PRL receptor, with lower affinity for the GH receptor [4]. Detection of hPL in maternal plasma occurs at approximately six weeks of gestation, and its concentration then increases linearly until about the thirtieth week of pregnancy, reaching peak concentrations of 5000–7000 ng/mL. The secretion rate of hPL near term is approximately 1 g/day, significantly greater than that of any other hormone [5]: indeed, the peak concentration of hPL is at least 25-fold that of PRL [6]. Maternal serum hPL levels are positively correlated with placental mass and are greater in multiple than singleton gestations [5]. hPL was widely used clinically in the 1970s–1980s, prior to widespread obstetric ultrasound, to assess fetoplacental wellbeing in late pregnancy [7,8]; but has since fallen from routine clinical use. As a pregnancy-specific hormone, hPL is rapidly washed from the maternal circulation following delivery of the placenta.

Along with estrogen, progesterone and PRL, hPL promotes third trimester mammary ductal and alveolar growth for lactogenesis (hence its designation as a ‘lactogenic’ hormone). However, hPL also has important metabolic roles in carbohydrate and lipid metabolism and fetal nutrient availability. It has been widely implicated in pregnancy-induced insulin resistance, maternal beta cell adaptation to pregnancy, and regulation of fetal growth in pre-clinical studies [3,9]. As such, altered hPL dynamics have been investigated in the context of metabolic conditions and outcomes in pregnancy.

The current literature on hPL in relation to maternal metabolism consists primarily of pre-clinical work and dated observational studies. Clinical research on hPL in human pregnancy has been relatively limited in recent decades, despite historical data suggesting that the hormone may have significant diagnostic and therapeutic potential.

In this systematic review, we examine current evidence regarding the relationship between hPL and maternal metabolic outcomes in pregnancy and postpartum, as well as key fetal outcomes, in the context of common metabolic conditions. We seek to provide mechanistic insights and examine the clinical implications of these findings.

## 2. Materials and Methods

### 2.1. Protocol and Registration

This review is part of a larger evidence synthesis examining lactogenic hormones in pregnancy and postpartum, and was conducted following the Preferred Reporting Items for Systematic Reviews and Meta-Analysis (PRISMA) 2020 guidelines. A protocol for this review is published [10] and was registered with the International Prospective Register of Systematic Reviews (PROSPERO), CRD42021262771.

### 2.2. Information Sources and Search Strategy

A systematic search strategy (Appendix A) combining MeSH terms and text words was developed using the OVID platform, in consultation with expert subject librarians, and translated to other databases. MEDLINE via OVID, MEDLINE ePub ahead of print, in-process, in-data review and other non-indexed citations via OVID, CINAHL plus, and Embase were searched on 8 July 2021 (updated 9 May 2022). Bibliographies of relevant studies identified by the search strategy were also manually searched to identify additional eligible studies.

### 2.3. Eligibility Criteria

Selection criteria using a modified version of the Participant, Exposure, Comparison, Outcome and Study Type (PECOT) framework [11] were established a priori. These were used to determine the eligibility of articles.

Studies were included in the systematic review when the following criteria were fulfilled: (i) participants were pregnant women and women up to 12 months postpartum; (ii) endogenous maternal serum hPL was measured and reported at least once during pregnancy; (iii) a comparison group of any type (or no comparison group) was reported; and (iv) one of the following key outcomes was reported in relation to hPL:

Maternal:Diabetes status during pregnancy and up to 12 months postpartum (pre-existing diabetes [type 1 or type 2], impaired glucose tolerance, or GDM; adequately defined)Metabolic indices (continuous measurements) related to maternal glucose/lipid metabolism (e.g., glucose measurements on oral glucose tolerance test; insulin secretion/sensitivity/resistance indices; beta-cell function) during pregnancy or up to 12 months postpartumObesity/body mass index, gestational weight gainPostpartum weight changePolycystic ovary syndromeLipid profile

Infant:Birthweight (absolute/centiles, macrosomia), growth restriction or placental mass in relation to pregnancies affected by maternal GDM or pre-gestational diabetes.

Eligible study types included cross-sectional, longitudinal cohort or case–control studies, and randomised controlled trials. Narrative and systematic reviews were excluded from the analysis, but their bibliographies were examined to identify relevant eligible articles. Commentaries, letters, conference abstracts, and case reports were excluded. Only full text English articles were included, with no date limits for eligibility. Maternal diabetes was considered adequately defined if the study clearly referred to type 1 or type 2 diabetes mellitus (T1DM, T2DM), GDM, or impaired glucose tolerance. If definitions were less clear (for example, older studies using White’s classification of diabetes in pregnancy), studies were included only if the information provided was sufficient to confidently deduce diabetes type. If definition adequacy varied between groups, the study was included only for the group(s) meeting definition requirements.

Studies were excluded if they were animal, in vitro or tissue/cell culture studies; involved exogenous administration of hPL; involved an intervention or procedure to manipulate hPL; focused on assisted reproductive technologies; or focused primarily on women with other pregnancy pathologies (e.g., pre-eclampsia, fetal death).

### 2.4. Study Selection and Risk of Bias Assessment

Two independent reviewers (KR and RG) screened all abstracts and full texts (Figure 1) and performed quality assessment, with 10% of studies assessed in duplicate. Quality appraisal (risk of bias) was conducted on Covidence software, using the Monash Centre for Health Research and Implementation (MCHRI) Evidence Synthesis Program tool [12], based on the Newcastle-Ottawa Scale for non-randomised studies [13]. Quality items were assessed using a descriptive component approach to evaluate external validity (study design, inclusion/exclusion criteria, and appropriateness of measured outcomes) and internal validity (selection, performance, detection, and reporting biases; attrition, confounding, statistical methodology, and study power). Studies that fulfilled all, most or few criteria were deemed to have low, moderate, or high risk of bias, respectively. Discrepancies were resolved through discussion and consensus.

### 2.5. Data Extraction and Synthesis

Data were manually extracted from all included studies by two independent reviewers using a purpose-built data extraction form in Microsoft Excel, with 10% extracted in duplicate. Information was collected on general study characteristics (authors, publication year and source, country, study design, duration of follow-up), participant characteristics (baseline age, metabolic conditions, parity, body mass index (BMI), ethnicity), hPL timepoints and values, hPL assay techniques, key maternal outcomes assessed in relation to hPL (unadjusted and adjusted, with consideration of covariates used), key relevant infant outcomes, and conclusions.

### 2.6. Evidence Synthesis and Statistical Analysis

Where meta-analysis was possible, Review Manager 5.4.1 software was used. Where published papers contained insufficient data to be entered into meta-analysis, details were sought from the authors. Weighted mean differences (WMD) were generated using random effects models. Statistical heterogeneity was assessed using the I^2^ test, with I^2^ values of >50% indicating moderate to high heterogeneity. Sensitivity analyses were conducted to examine the effects of studies with high risk of bias on the overall results. Sensitivity analysis was also performed with exclusion of studies published prior to the year 2000 (given that older studies likely reflect a different clinical and therapeutic environment). Where meta-analysis was not possible, narrative synthesis of results was performed. Data are presented in summary tables and in narrative format. Forest plots (Figure A1 and Figure A2a,b) and funnel plots (Appendix A) were used to present the results of meta-analyses and publication bias assessments, respectively.

## 3. Results

### 3.1. Study Selection and Characteristics

A total of 3922 results were retrieved from the initial database search for the broader evidence synthesis examining lactogenic hormones. Following removal of duplicates, 2643 and 190 studies were excluded at abstract and full text screening, respectively (Figure 1). Of note, studies excluded on the basis of unavailable English full text (*n* = 51) or inadequate maternal diabetes definition (*n* = 51) were disproportionately dated, with all published prior to 1998 and 2000, respectively.

A total of 62 studies met the broader eligibility criteria for inclusion, of which 35 pertained specifically to hPL and were included in the current review. Due to methodological heterogeneity, meta-analysis was only possible for hPL differences in late pregnancy by T1DM status (four studies) and for hPL differences in early and late pregnancy by GDM status (three and 10 studies, respectively).

### 3.2. Risk of Bias of Included Studies

Of the 35 included studies, 12 were deemed high risk of bias, 16 moderate, and seven low (Table A1, Table A2, Table A3, Table A4 and Table A5). The main aspects contributing to high risk of bias were the presence of confounding and selection bias, both of which were present in seven of the 12 studies deemed high risk of bias. Low-quality statistical analysis and concerns regarding the accuracy and validity of hormone measurement were also common domains of concern, each present in six of the 12 high-risk studies.

Based on visual inspection of funnel plots, there was no evidence of publication bias across any of the outcomes assessed (Appendix A).

### 3.3. Synthesis of Results

#### 3.3.1. Human Placental Lactogen in Pregnancies Affected by Pre-Gestational Metabolic Conditions

Twelve studies examined hPL across pregnancy in women with adequately-defined pre-gestational diabetes mellitus (PGDM) (Table A1). These were all published prior to 1998 and focused on T1DM, with no studies in pregnancies affected by T2DM or PCOS.

(a)Differences in hPL between T1DM and control pregnancies

Seven of the 12 studies [14,15,16,17,18,19,20] reported measuring early pregnancy (≤24 weeks) hPL in women with T1DM compared with controls, although only six clearly reported between-group hPL results. Of these, four found hPL to be significantly lower in T1DM than controls for at least one early pregnancy timepoint [14,15,16,17]. One found hPL to be higher in T1DM than controls [18], while the other found no difference [20]. With the exception of one study [16], all of these studies lacked sufficient raw published data for meta-analysis, and their age precluded contacting authors for more details.

Nine studies compared late pregnancy (>24 weeks) hPL between T1DM and controls. Of these, four [16,18,21,22] had available data for meta-analysis (Figure A1). Latest available pregnancy measurements were used in all cases (all 34–40 weeks). Pooled results showed significantly higher late pregnancy hPL levels in women with T1DM than controls (WMD = 1.24 µg/mL, 95% CI 0.44 to 2.05, *p* = 0.003), with no heterogeneity (I^2^ = 0%, *p* = 0.6). Sensitivity analysis with exclusion of the two studies deemed high risk of bias did not significantly alter results. Of the five studies without sufficient information for inclusion in the meta-analysis; two showed significantly higher late hPL levels in T1DM than control pregnancies [19,23], two showed no difference [15,20], and a single small study showed lower late hPL in T1DM than controls [14].

Two studies which sub-divided T1DM subjects by class (White’s classification, based on complications and duration) found no difference in hPL values across subgroups of progressive T1DM ‘severity’ [17,18]. However, a third study (focused specifically on diabetic retinopathy) found higher hPL values in the subset of pregnant women with T1DM with retinopathy, particularly progressive retinopathy, who also had worse glycaemic control and higher placental mass [19].

(b)Relationship between hPL and glycaemic measures in T1DM

Five studies [14,15,18,21,22] examined hPL in T1DM in relation to plasma glucose (mean and/or prevailing). Botta et al. [14] reported that hPL was inversely associated with blood glucose levels across gestation (both average glucose that day and glucose at the time of hPL sampling). The remaining studies, of which two had similar methodology [15,18] and two sampled hPL serially over 8–24 h [21,22] reported no relationship between hPL and glucose.

Two studies examined HbA1c in relation to hPL in T1DM (one in a one-off early pregnancy sample, one at serial timepoints); both showing no significant relationship [15,17].

Three studies examined hPL levels relative to the increase in insulin requirements across pregnancy in T1DM, all reporting no relationship between these two variables [18,20,24]. In contrast, the sole study to use insulin clamp methodology to directly quantify insulin resistance in early and late pregnancy found that the size of hPL increment across pregnancy was significantly inversely proportional to late pregnancy insulin sensitivity in women with T1DM (*n* = 6) [25].

#### 3.3.2. Human Placental Lactogen in Pregnancies Affected by Gestational Diabetes Mellitus

Seventeen studies examined hPL across pregnancy in women with GDM (Table A2).

(a)Differences in hPL between GDM and control pregnancies

Five studies [16,18,26,27,28] compared hPL in women with GDM and controls in early pregnancy (≤24 weeks, often prior to GDM diagnosis and recognition). In the three studies with sufficient data for meta-analysis (Figure A2a), pooled analysis showed no significant difference in early pregnancy hPL between GDM and control pregnancies (WMD = 0.21 µg/mL, 95% CI −0.52 to 0.94, *p* = 0.6). Statistical heterogeneity was high (I^2^ = 74%, *p* = 0.02), and small sample sizes universal. Two of the three studies [16,26] were deemed high risk of bias; precluding sensitivity analysis. Of the two studies with insufficient detail for inclusion in the meta-analysis, one reported higher hPL values in women with GDM than controls [28], and the other showed no significant difference herein [27].

Fourteen studies compared hPL between women with GDM and controls in later pregnancy (>24 weeks), twelve of which were eligible for meta-analysis, but two were subsequently excluded due to significant methodological concerns and a suspicion of erroneous hPL values (see Table A2 footnotes) [29,30]. All values were third trimester samples. Pooled analysis of the ten included studies [16,18,22,26,31,32,33,34,35,36] using the latest timepoint if multiple were available (Figure A2b), suggested no significant difference in late pregnancy hPL between women with GDM and controls (WMD = 0.47 µg/mL, 95% CI −0.14 to 1.09, *p* = 0.1), with moderate heterogeneity (I^2^ = 60%, *p* = 0.008). Sensitivity analyses with exclusion of older studies and those deemed high risk of bias did not significantly alter results. The two studies that lacked sufficient detail for inclusion in the meta-analysis also found no significant difference in late hPL between GDM and controls [37,38].

Three studies examined the clinical utility of hPL as a risk predictor for GDM. Two studies (one with major methodological limitations, see Table A2 footnotes [29]) suggested it was unlikely to be useful, due to poor classification performance [28] or non-significant predictive capacity [29]. Conversely, the third study [34] suggested that hPL may be a promising adjunct to screening glucose challenge tests in predicting the likelihood of a subsequent abnormal oral glucose tolerance test (OGTT).

(b)Relationship between hPL and glycaemic measures in GDM

Nine studies examined hPL in relation to cross-sectional or longitudinal glycaemic parameters in GDM such as plasma glucose or insulin, or markers of insulin sensitivity or resistance. Overall, none found consistent relationships between hPL and these variables in women with GDM or controls [18,22,29,30,32,33,36,38,39].

#### 3.3.3. Human Placental Lactogen in Relation to Glycaemic or Insulin-Related Parameters in Healthy Pregnancies and Postpartum

Four studies [40,41,42,43] examined hPL in relation to glycaemic or insulin-related parameters in healthy pregnant women (Table A3). The methodology of these studies varied considerably, precluding meta-analysis. Benny et al. [40] examined hPL dynamics across a 24 h period in the third trimester, showing a peak after overnight fasting (temporally coincident with the time of lowest glucose and insulin, and potentially consistent with the idea of hPL as an insulin-antagonistic hormone). Enzi et al. [41] found that maternal hPL levels at 34 weeks were positively related to the area under the curve (AUC) of both glucose and insulin, suggesting this confirmed the diabetogenic effects of hPL. This differed from the findings of two other studies, where hPL was unrelated to prevailing glucose [42] or to 2 h OGTT insulin or glucose levels [43] in healthy pregnancies. However, higher hPL levels were associated with higher levels of non-esterified fatty acids in one study [42], suggestive of anti-insulin, diabetogenic properties.

One study [44] related hPL to postpartum glycaemia, finding that hPL in late pregnancy was not an independent predictor of insulin resistance, beta-cell function or diabetes risk (all measured at 3 months postpartum).

#### 3.3.4. Human Placental Lactogen in Relation to Body Mass Index or Gestational Weight Gain in Pregnancy

Four studies [30,41,45,46] examined hPL in relation to maternal BMI or gestational weight gain (GWG) (Table A4). Two studies showed no relationship between hPL and maternal BMI [30] or hPL and GWG (crudely categorised as <20% or >20% ideal body weight for normal or excessive GWG, respectively) [41]. Lin et al. [45] described an inverse relationship between hPL and absolute maternal weight at term, proposed to be a dilutional effect (more tissue space in larger women). McCarrick et al. [46] found that obese women were over-represented in a group of women with low hPL but normal estrogen levels, suggesting that obesity may impact on hPL regulation and activity (although many of these women had other pregnancy complications which may have explained their low hPL levels, such as toxaemia or intra-uterine growth restriction).

#### 3.3.5. Human Placental Lactogen in Relation to Fetal, Neonatal or Placental Outcomes in Pregnancies Affected by Maternal Diabetes

Seven studies [14,18,20,35,38,47,48] examined hPL in relation to fetal, neonatal or placental outcomes in pregnancies affected by maternal PGDM/GDM (Table A5). Variable methodology and lack of reporting detail prevented meta-analysis.

One study [48] examined hPL in relation to fetal growth/size in the early stages of T1DM pregnancies (*n* = 26), and found that hPL at 7–16 weeks could be best related to menstrual age when the latter was corrected by any ultrasonographically determined ‘growth delay’. Given that hPL reflects functional trophoblastic mass, this suggested that the observed growth delay in T1DM pregnancies may relate to delayed placental development.

Three studies examined hPL in the late third trimester relative to placental weight at delivery in pregnancy cohorts affected by (adequately-defined) maternal PGDM/GDM. One study [18] of 38 women found that late pregnancy hPL was strongly positively correlated to placental weight in T1DM (r = 0.8, *p* < 0.01), GDM (r = 0.6, *p* < 0.05), and controls (r = 0.6, *p* < 0.05). In the remaining two studies, one found no relationship (despite a trend noted) between late pregnancy hPL and placental weight in a small combined cohort of 15 T1DM and 10 control pregnancies [14] and the other found that hPL was positively associated with placental mass in the larger control group (*n* = 69; r = 0.3, *p* < 0.01) but not in the T1DM cohort (*n* = 40), likely due to low statistical power [20].

Five studies examined hPL in relation to infant birthweight in pregnancies affected by PGDM/GDM. Two [14,35] found no relationship between hPL at 36 weeks or at term with birthweight in a combined cohort of women with T1DM and controls (*n* = 25) or a combined cohort of women with GDM, women with premature deliveries and controls (*n* = 46), respectively. Conversely, two other studies showed positive relationships between third trimester hPL and corrected birthweight in T1DM (r = 0.48, *p* < 0.02) [20] or birthweight in a GDM cohort (r = 0.59, *p* < 0.05) [38]. Finally, Small et al. [47] examined hPL in relation to birthweight ‘class’, finding that a T1DM group with macrosomic infants (mean birthweight 3.96 kg at 37 weeks) had significantly higher hPL at 34 weeks than matched T1DM pregnancies without macrosomia (mean birthweight 3.05 kg at 37 weeks).

## 4. Discussion

To our knowledge, this is the first systematic review of hPL in relation to maternal metabolic outcomes in pregnancy. Specifically, we explored hPL in healthy pregnancies and in those with PGDM/GDM, and relationships to maternal metabolic parameters and fetal growth within these subgroups. Systematic review and meta-analysis suggests altered hPL dynamics in pregnancies affected by T1DM, but no relationships with GDM were identified. hPL appears positively correlated with placental mass in PGDM/GDM and elevated in pregnancies affected by macrosomia. However, hPL levels were not clearly linked to maternal glycaemic outcomes in PGDM/GDM, despite pre-clinical evidence for physiological roles in both insulin resistance and maternal beta-cell adaptation to pregnancy.

### 4.1. hPL in Pre-Gestational (Type 1) Diabetes Mellitus

In pre-gestational T1DM, the results of our review suggest altered hPL dynamics across gestation (with differential effects in early and late pregnancy). Meta-analysis comparing early hPL levels in T1DM vs. control pregnancies was not possible due to a lack of detailed comparative data, but results were broadly suggestive of lower early hPL levels in pregnancies affected by T1DM. Whilst some authors have speculated that these low levels may be a direct response to maternal hyperglycaemia [14], it should be noted that experimental evidence showing depression of hPL levels required maternal blood glucose to be raised dramatically (via rapid intravenous infusion over 30 min to a mean of 22.2 mmol/L, in the seminal trial) [49]. The evidence summarised in our review suggests that more subtle alterations of plasma glucose, such as may occur in adequately-controlled maternal T1DM, are unlikely to have a major direct impact on hPL levels. Overall, the lower levels of hPL observed in early T1DM pregnancy seem more likely to relate to delayed trophoblastic development [16,48]. Such a mechanism would be consistent with the observation that concentrations of other key gestational hormones–such as PRL and human chorionic gonadotropin (hCG)–may also lag behind normal early pregnancy reference ranges in T1DM pregnancies, particularly in the context of suboptimal glycaemic control [50].

In the later part of T1DM pregnancy (namely the third trimester), the results of our review and meta-analysis support higher circulating maternal hPL levels in T1DM than control pregnancies. The observation of higher hPL levels in T1DM has typically been attributed to greater placental mass in such pregnancies. Mechanistically, this may reflect fetal hyperglycaemia with secondary hyperinsulinaemia, leading to macrosomic stimulation of the placenta [8]. Early clinical literature, dating to the era where the hormone was in routine obstetric use, is in keeping with this: high-normal or high hPL levels were “expected” in T1DM pregnancy. When levels fell below this, they were likely to be suggestive of a separate superimposed reason for fetoplacental compromise (such as toxaemia or late fetal demise, which were common occurrences in T1DM cohorts in that era) [51].

Together, the results of studies of hPL in T1DM suggest that the relationship between hPL and maternal metabolism is likely bidirectional. Whilst hPL certainly has metabolic actions, its concentrations are also likely influenced by an altered maternal metabolic environment (such as in T1DM), with different mechanisms operational in early vs. late pregnancy.

### 4.2. hPL in Maternal Glycaemia and Gestational Diabetes Mellitus

Our meta-analysis of studies comparing absolute hPL concentrations between women with GDM and controls in both early and late pregnancy showed no statistically significant differences between groups. Similarly, the small number of studies which investigated hPL as a GDM risk prediction biomarker do not support its predictive utility.

A body of pre-clinical evidence certainly provides theoretical grounds to suggest that hPL may be immediately relevant to maternal glucoregulation in pregnancy: at high concentrations, hPL has classically been considered a ‘diabetogenic’ hormone [8,52] with insulin-antagonistic and lipolytic effects. Most endocrine texts still describe hPL as a key contributor to gestational insulin resistance, increasing fetal nutrient availability by sparing glucose, amino acids and ketones for placental-fetal transport [53]. However, rodent and in vitro human data have also repeatedly identified a key parallel role for hPL (acting via the PRL receptor) to induce maternal pancreatic adaptation to pregnancy, increase beta-cell mass, and potentiate glucose-stimulated insulin secretion [6,54,55].

Despite these roles, the human data synthesised in our review suggests that absolute maternal hPL concentrations measured in human pregnancy populations may be difficult to link directly to glycaemic parameters. As such, it seems likely that the in vivo metabolic effects of hPL are likely to be much more complex than suggested by existing pre-clinical evidence, much of which was accumulated a generation ago. For example, there is increasing acknowledgment of the multifactorial and synergistic nature of late pregnancy insulin resistance, with important roles for GH-V (a powerful lipolytic hormone), maternal insulin like growth factor 1 (IGF-1), progesterone, cortisol and tumor necrosis factor (TNFα); as well as a fall in adiponectin [9]. As such, the designation of hPL as the “major diabetogenic stress factor of pregnancy” may be overly simplistic. Similarly, autopsy evidence from the pancreata of pregnant women indicates that the adaptive beta cell changes of human pregnancy may be less profound and different in nature to those observed in rodents [56], which immediately suggests that the extrapolation of findings from sub-primate models about the insulinogenic properties of the hormone must be approached with caution. Furthermore, circulating serum levels of a hormone do not always tell the whole story: for example, recent work has suggested that certain PRL receptor polymorphisms may predict GDM risk, implying that differences in hormone action—at a tissue level—may be just as important as absolute hormone concentrations [57].

Thus, whilst pre-clinical studies clearly support a key role for hPL in metabolic adaptations to human pregnancy (both as an insulin antagonist and as a stimulus for augmented insulin secretion); the collated observational data suggest that its measured circulating concentrations may not provide direct insights into maternal glucose homeostasis.

### 4.3. hPL in Fetal Growth in Pregnancies Affected by Maternal Diabetes

Studies have consistently demonstrated a positive association between hPL and placental mass (in both diabetic and non-diabetic cohorts) [14,18,51,58,59] and a positive association between hPL and infant birthweight has also been demonstrated in several large general pregnancy cohorts [60,61,62]. Our review, which was limited to pregnancies affected by adequately-defined maternal diabetes, generally supported these findings. Accurate antenatal prediction of fetal macrosomia remains challenging, and current strategies (including fundal measurements and ultrasound assessment) are resource-intensive. There is thus a clear requirement for maternal serum biomarkers in improving macrosomia prediction, particularly in women at high risk (such as those with PGDM/GDM). Whilst several biomarkers have been assessed for their association with birthweight or macrosomia (both in diabetic and non-diabetic pregnancies), evidence is mixed and uncertainties around clinical utility persist [63]. hPL has recently been largely overlooked in this capacity, but previous work suggests it may have significant potential if revisited [19,47].

Mechanistically, a direct role for hPL in the regulation of fetal growth is also feasible: for example, targeted reductions in placental lactogens in sheep pregnancy via modification of placental gene expression result in significantly reduced fetal weight, possibly mediated by disrupted IGF-1 and IGF-2 expression [64,65]. In humans, low levels of hPL in small for gestational age pregnancies are commonly observed (along with reduced levels of GH-V) [62,66,67]. The role of hPL in large for gestational age (LGA) pregnancies—particularly those affected by maternal metabolic disease—is similarly interesting. In general obstetric populations, significant positive relationships between maternal hPL levels at 34 weeks’ gestation and neonatal body weight, body fat mass and fat cell weight have been reported [41], and other research has demonstrated a 1.6-fold higher expression of hPL genes in the placentas of LGA newborns compared to those of normal size [67]. As such, hPL may contribute aetiologically to macrosomia, aside from simply reflecting increased placental mass in LGA pregnancies. Whilst fetal overgrowth in maternal obesity and diabetes is commonly associated with placentomegaly, it is also possible that the resulting hPL excess may further stimulate both maternal and fetal beta-cell expansion and increase fetal insulin production, which would promote glycogenesis, fat deposition and fetal growth [9].

Given the likely positive relationships between hPL and placental mass/neonatal weight in pregnancies affected by maternal diabetes, as well as a possible aetiological role in the development of macrosomia; late-pregnancy hPL warrants re-visiting in modern obstetric populations (both with and without diabetes).

### 4.4. Strengths and Limitations

As noted above, this is the first review to systematically collate and synthesise the literature linking hPL to maternal metabolic outcomes in pregnancy and related fetal outcomes. We employed rigorous, international gold-standard methodology with a protocol developed a priori to ensure transparency. The review addresses a broad, mechanistic question that links important aspects of female reproductive and metabolic health; and sheds light on a hormone which has been overlooked in the endocrine literature in recent decades. Identification of biomarkers that may aid with GDM risk prediction, or help with identifying complications in pregnancies affected by maternal diabetes; is a key health priority–particularly given the accumulating body of evidence linking gestational metabolic disease (and insulin resistance) to lifetime metabolic and cardiovascular risk in women.

Limitations of the review process include restriction of the search to published English language articles. In addition, the requirement for clearly defined maternal diabetes type excluded some older studies (pre-1980s, often referring only to ‘maternal diabetes’). Inclusion of this literature would have increased the number of included studies–and possibly numbers for meta-analysis–but would have introduced significant uncertainty and made results less applicable to modern clinical populations.

Limitations of the literature were substantial and precluded firm conclusions regarding the role of hPL in pregnancy and postpartum in the context of common metabolic conditions. These limitations included heterogeneous methodology and a frequent lack of detail in data reporting, which contributed to the inability to perform meta-analysis for many outcomes. Variable study quality was reflected in the risk of bias assessments (28 of 35 studies were deemed to have moderate or high risk of bias). Studies were small and were all observational in nature, increasing the likelihood of low statistical power and residual confounding. Measurement of hPL at only one or two timepoints (and often within a broad gestational age bracket, without subsequent correction for exact gestational age) was a significant limitation of many studies, given the steep increase in hPL concentrations known to occur across normal pregnancy. BMI is also an important potential confounder in the relationship between hPL and metabolic indices, but BMI reporting in the included studies was variable, and precluded stratification of meta-analyses based on BMI. Data (on T1DM, in particular) was dated; and thus reflected a historical therapeutic environment. Assay methodology for measuring hPL also varied, with older studies using radioimmunoassay techniques and newer studies favouring enzyme-linked immunoassays. There were no data relating hPL to maternal metabolic outcomes or fetal parameters in T2DM or PCOS cohorts, or to lipid profiles; and data on maternal obesity and GWG was sparse. Finally, the hormonal environment of pregnancy and postpartum is complex, and studies focusing on absolute levels of a single hormone may overlook other factors such as hormone synergy, local tissue levels, and receptor polymorphisms.

## 5. Conclusions

In summary, the findings of our review suggest that in T1DM pregnancies, hPL levels may be lower than controls in early pregnancy (possibly reflecting delayed placental development) and higher than controls in later pregnancy (likely in keeping with higher placental masses), but that absolute hPL concentrations are not clearly linked to maternal glycaemic outcomes in PGDM or GDM, nor to GDM status/risk. Moreover, hPL is likely positively related to placental mass and infant birthweight in pregnancies affected by PGDM or GDM, and may be aetiologically important in the regulation of fetal growth. Despite having fallen from routine clinical use in recent decades, hPL may warrant renewed investigation as an antenatal biomarker for the prediction of macrosomia. However, given the limited available data, small study numbers, and substantial heterogeneity in study design and methodology, future high-quality studies exploring this hormone in well-defined contemporary populations are required to clarify these relationships and to inform future research and clinical practice.

## Figures and Tables

**Figure 1 ijms-23-15621-f001:**
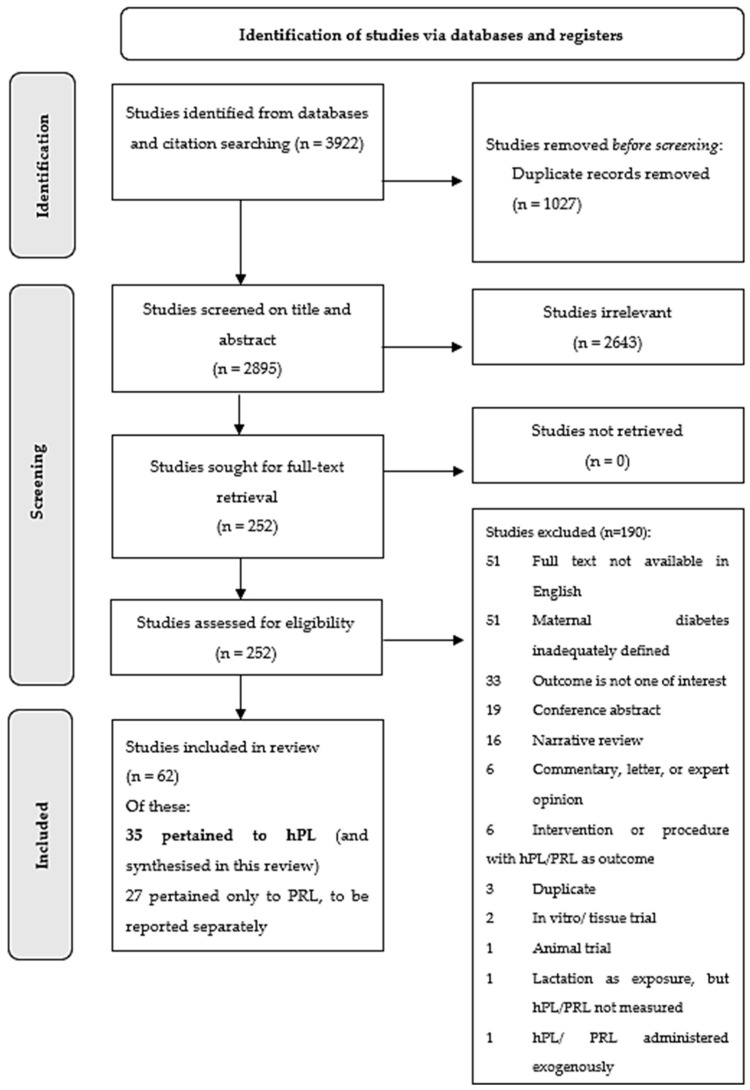
PRISMA flowchart.

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
