# Peer review of "Human Placental Lactogen in Relation to Maternal Metabolic Health and Fetal Outcomes: A Systematic Review and Meta-Analysis"

_ijms, 2022, doi:10.3390/ijms232415621_

Round 1

Reviewer 1 Report

Summary

The manuscript by Kate Rassie etal is the first systematic review and meta-analysis evaluating associations between human placental lactogen (hPL) and maternal or fetal outcomes. This paper is well written, the protocol has been published and registered. However, the results cannot fully support the conclusion, lack of detailed and comprehensive analysis.

Major and Minor marks

As follows from the manuscript, authors aim to charity the links between levels of hPL and maternal or fetal outcomes, the introduction should focus on the importance of hPL, the research status of hPL and maternal disease with obesity, pre-gestational diabetes (type 1 or 2), or PCOS, rather than emphasizing the particularities of maternal pregnancy, which is well-known.

About results, authors examine current evidence and mainly analyzed the differences in the distribution of hPL in maternal states (type 1, GDM). Whether a BMI stratified analysis should be performed considering the influence of body weight on glucose metabolism or islet function in GDM. In addition, researchers also paid attention to the relationship between concentrations of hPL and metabolic indicators (HbA1c, insulin, insulin resistance index, BMI), but only showed the results, and meta-analysis of the correlation should be conducted. As well as in the part of that hPL relation to fetal, neonatal or placental outcomes, three studies examined hPL in the late third trimester relative to placental weight and five studies examined hPL in relation to infant birthweight. The corresponding analysis results should be added. Finally, the authors should pay attention to the presentation of graphics. 

Author Response

Reviewer 1 responses

Summary: The manuscript by Kate Rassie et al is the first systematic review and meta-analysis evaluating associations between human placental lactogen (hPL) and maternal or fetal outcomes. This paper is well written, the protocol has been published and registered. However, the results cannot fully support the conclusion, lack of detailed and comprehensive analysis.

Major and Minor marks

As follows from the manuscript, authors aim to clarify the links between levels of hPL and maternal or fetal outcomes, the introduction should focus on the importance of hPL, the research status of hPL and maternal disease with obesity, pre-gestational diabetes (type 1 or 2), or PCOS, rather than emphasizing the particularities of maternal pregnancy, which is well-known.

We thank the reviewer for their time and meaningful feedback. Regarding the importance of hPL in relation to maternal metabolic health, the key points have been described in the introduction.

Our introduction deliberately begins with a general description of glycaemia and metabolism in pregnancy, after which we introduce human placental lactogen and describe its importance and role in normal human pregnancy. We would maintain that both of the aforementioned elements are necessary for the paper to be accessible to a general scientific audience (and as a basis for the ensuing discussion of hPL in metabolically abnormal pregnancies).

Lines 73-81 of the introduction then describe the potential relevance of hPL in maternal metabolic disease (obesity, pre-gestational and gestational diabetes, PCOS), and address the research status of hPL in this context; providing background and rationale for the aim of the current review (presented in the final section of the introduction, lines 82-85).

About results, authors examine current evidence and mainly analyzed the differences in the distribution of hPL in maternal states (type 1, GDM). Whether a BMI stratified analysis should be performed considering the influence of body weight on glucose metabolism or islet function in GDM. In addition, researchers also paid attention to the relationship between concentrations of hPL and metabolic indicators (HbA1c, insulin, insulin resistance index, BMI), but only showed the results, and meta-analysis of the correlation should be conducted. As well as in the part of that hPL relation to fetal, neonatal or placental outcomes, three studies examined hPL in the late third trimester relative to placental weight and five studies examined hPL in relation to infant birthweight. The corresponding analysis results should be added.

The reviewer makes valid suggestions and we agree that these analyses would have been useful. Unfortunately, these additional meta-analytic components were not feasible for the following reasons:

  • Within studies of hPL in relation to GDM, reporting of maternal BMI was incomplete and heterogenous, often with a wide BMI range for study eligibility and no breakdown of BMI according to either maternal diagnosis and/ or hPL levels. BMI stratification in the meta-analysis of hPL according to GDM status was thus not possible.
  • Meta-analysis was not possible for the relationships between hPL and metabolic indices (such as HbA1c, insulin, IR, BMI) due to small study numbers and marked heterogeneity in study design/ methodology. This limited the presentation of results to the (current) tabulation and narrative synthesis (this has now been acknowledged in lines 309-310 of the results section). The same applied for the placental and neonatal weight outcomes, in which methodology and maternal condition varied widely between studies. Most studies reported results in insufficient detail for inclusion in meta-analysis (e.g. graphical representation only, and/ or a single correlation coefficient), and their dated nature precluded contacting of the authors for raw data. The inability to perform meta-analysis has now been explicitly acknowledged in lines 338-339.

Overall, the authors have completed meta-analysis for all outcomes in which studies were sufficiently homogenous to enable meaningful conclusions to be drawn. We do acknowledge the inability to perform meta-analysis for all outcomes of interest as a limitation of our review process, and have mentioned this in the limitations section of the manuscript (lines 497-499).

Finally, the authors should pay attention to the presentation of graphics. 

The figures in the manuscript are in accordance with the IJMS formatting template and no issues were flagged by the other reviewer or the editors. We are happy to address specific concerns around the presentation of graphics, should the editorial office feel that this is necessary.

Reviewer 2 Report

In this systematic review manuscript, the authors aim to examine the current evidence regarding the relationship between hPL and maternal metabolic outcomes in pregnancy and postpartum, as well as key fetal outcomes, in the context of common metabolic conditions. This is a good systematic review and I only have some minor suggestions. 

1.    Apart from the quantitative methods, some qualitative methods to assess the risk of biasness of the selected articles can also be employed, such as evaluating the selection bias, performance bias, detection bias, and attrition bias.

2.    Countries of the experiments conducted in the articles should also be listed in Table A1 as there may be differences in results between patients from different country of origin.

3.    Minor text editing and proofreading.

Author Response

Reviewer 2 responses

In this systematic review manuscript, the authors aim to examine the current evidence regarding the relationship between hPL and maternal metabolic outcomes in pregnancy and postpartum, as well as key fetal outcomes, in the context of common metabolic conditions. This is a good systematic review and I only have some minor suggestions. 

  1. Apart from the quantitative methods, some qualitative methods to assess the risk of biasness of the selected articles can also be employed, such as evaluating the selection bias, performance bias, detection bias, and attrition bias.

We thank the reviewer for their positive feedback. All articles indeed underwent a qualitative, itemised risk of bias assessment against all the domains suggested by the reviewer, using the MCHRI critical appraisal template which is referenced in the methodology section (lines 142-151) and is based on the Cochrane risk of bias methodology. However, with 35 included studies assessed against 22 quality components by 2 reviewers, tabulation of all risk of bias aspects for all articles was too lengthy for inclusion in the final publication.

The overall risk of bias score for each article has been tabulated in the far right column of Tables A1-A5. The condensed results of the qualitative assessments specifically (including the main domains contributing to high risk of bias) are also summarised in the Results section, lines 217-222.

  1. Countries of the experiments conducted in the articles should also be listed in Table A1 as there may be differences in results between patients from different country of origin.

We agree with this suggestion. Countries of origin for each study have now been included in all tables.

  1. Minor text editing and proofreading.

The manuscript has been re-read and proofread again, with any detected typographical errors amended. We are happy to address any further issues specifically requested by the reviewers or editors.

Round 2

Reviewer 1 Report

The manuscript could not complete the revision according to the comments.

Firstly, authors should simply the 1st para and 2nd para in the context of introduction, and spend a lot of space to focus on the essentials.

Secondly, of particular importance, as the author puts it, “these analyses would have been useful”, which were in agree with the revision on results. If the study has these problems, likely insufficient data, small study numbers and marked heterogeneity in study design/ methodology, data collection should continue and use own demographic data to make up for it.

Overall, insufficient evidence with unreliable results, whether conclusions were not sufficiently based on scientific evidence, or whether the study presented questionable validity. Moreover, this is currently no suitable to guide the clinical decision on rechallenge.

Author Response

Reviewer Comment: The manuscript could not complete the revision according to the comments. Firstly, authors should simply the 1st para and 2nd para in the context of introduction, and spend a lot of space to focus on the essentials. Secondly, of particular importance, as the author puts it, “these analyses would have been useful”, which were in agree with the revision on results. If the study has these problems, likely insufficient data, small study numbers and marked heterogeneity in study design/ methodology, data collection should continue and use own demographic data to make up for it. Overall, insufficient evidence with unreliable results, whether conclusions were not sufficiently based on scientific evidence, or whether the study presented questionable validity. Moreover, this is currently no suitable to guide the clinical decision on rechallenge.

Author response: We thank the reviewer for their feedback and we agree that small sample sizes, heterogeneous study methodology, and inadequately-detailed reporting are key limitations of the included articles. However, this reflects the overall state of the literature and is thus not amenable to modification by the authors. We have elaborated on this in the manuscript at some length (lines 499 – 512) and have now clarified this further in the strengths and limitations (page 11-12) and in the conclusions (page 12) sections of the manuscript.

Notwithstanding these limitations, we maintain that the evidence synthesis presented herein represents a valuable contribution to the field, summarising the current state of the literature and highlighting key priority areas for future exploration. Further research in this area, such as the currently funded biomarker studies planned by the authors, must be guided by current gaps in the literature in order to add value and advance scientific understanding.

Round 3
